# Anticancer Therapy with HDAC Inhibitors: Mechanism-Based Combination Strategies and Future Perspectives

**DOI:** 10.3390/cancers13040634

**Published:** 2021-02-05

**Authors:** Robert Jenke, Nina Reßing, Finn K. Hansen, Achim Aigner, Thomas Büch

**Affiliations:** 1University Cancer Center Leipzig (UCCL), University Hospital Leipzig, D-04103 Leipzig, Germany; 2Clinical Pharmacology, Rudolf-Boehm-Institute for Pharmacology and Toxicology, Medical Faculty, University of Leipzig, D-04107 Leipzig, Germany; thomas.buech@medizin.uni-leipzig.de; 3Department of Pharmaceutical and Cell Biological Chemistry, Pharmaceutical Institute, Rheinische Fried-rich-Wilhelms-Universität Bonn, D-53121 Bonn, Germany; nressing@uni-bonn.de (N.R.); finn.hansen@uni-bonn.de (F.K.H.)

**Keywords:** histone deacetylase, histone deacetylase inhibitor, cancer, bifunctional inhibitors, PROTAC

## Abstract

**Simple Summary:**

Beyond mutations, epigenetic changes have been described as drivers for cancer as well. While leaving the overall DNA structure intact, they can be responsible for tumor malignancy by mediating the transcriptional upregulation of oncogenes. This provides the basis for “epigenetic therapies” in cancer. Histone deacetylases (HDACs) are major players in epigenetic reprogramming. HDAC inhibitors (HDACis), either with broad-spectrum activity on various HDAC isoforms or with specific subtype specificity, have shown promising anticancer efficacies. The tremendous number of genes potentially affected creates the possibility for the parallel targeting of multiple disease-relevant pathways. Here, we give a comprehensive overview of various preclinical and clinical studies on HDACis. A particular focus is placed on the detailed description of promising strategies based on the combination of HDACis with other drugs. This also includes the development of new bifunctional inhibitors as well as novel approaches for HDAC degradation, rather than inhibition, via PROteolysis-TArgeting Chimeras (PROTACs).

**Abstract:**

The increasing knowledge of molecular drivers of tumorigenesis has fueled targeted cancer therapies based on specific inhibitors. Beyond “classic” oncogene inhibitors, epigenetic therapy is an emerging field. Epigenetic alterations can occur at any time during cancer progression, altering the structure of the chromatin, the accessibility for transcription factors and thus the transcription of genes. They rely on post-translational histone modifications, particularly the acetylation of histone lysine residues, and are determined by the inverse action of histone acetyltransferases (HATs) and histone deacetylases (HDACs). Importantly, HDACs are often aberrantly overexpressed, predominantly leading to the transcriptional repression of tumor suppressor genes. Thus, histone deacetylase inhibitors (HDACis) are powerful drugs, with some already approved for certain hematological cancers. Albeit HDACis show activity in solid tumors as well, further refinement and the development of novel drugs are needed. This review describes the capability of HDACis to influence various pathways and, based on this knowledge, gives a comprehensive overview of various preclinical and clinical studies on solid tumors. A particular focus is placed on strategies for achieving higher efficacy by combination therapies, including phosphoinositide 3-kinase (PI3K)-EGFR inhibitors and hormone- or immunotherapy. This also includes new bifunctional inhibitors as well as novel approaches for HDAC degradation via PROteolysis-TArgeting Chimeras (PROTACs).

## 1. Introduction

### 1.1. Histone Acetylation and Regulation of Gene Expression

Cancer is often considered as a disease driven by the mutation of genes involved in growth or differentiation—in particular, so-called tumor oncogenes or suppressor genes [1,2]. Yet, there exists a powerful machinery to drive malignancies without affecting the DNA structure itself [3]. So-called “epigenetic” alterations alter the structure of the chromatin and consequently influence the transcription of genes. They can occur at any time during cancer progression but have also been described in other processes such as, for example, the initiating step in malignant degeneration, thus emphasizing their overall relevance [3]. Human cells contain the genetic code as chromatin. Nucleosomes display the basic units composed of 147 bp DNA wrapped around an octamer of the four histones H2A, H2B, H3 and H4 [4,5,6,7]. The N-terminal tails of the core histone contain many lysine and arginine residues, which makes them prone to post-translational modifications [8,9].

A wide variety of possible modifications exists, including, for example, methylation, phosphorylation, sumoylation, deamination and, importantly, acetylation. Histone acetylation was described as early as 1964 and already then considered to affect RNA synthesis [10]. Acetylation of core histones neutralizes the positive charge of the lysine residue, thereby weaking their interactions with the negatively charged DNA molecule. In fact, two different states of chromatin exist. Euchromatin represents the “open”, mostly acetylated state, which leads to unfolded DNA and access for transcription factors. In contrast, heterochromatin is largely hypoacetylated, resulting in a tightly condensed DNA less accessible for transcription and thus considered as “silent” [11]. Acetylation is a reversable mechanism, with transition from one state to the other catalyzed by histone acetyltransferases (HATs) on the one hand and histone deacetylases (HDACs) on the other. While the HATs can be divided into three major families, HDACs include four families with 18 different HDACs [5,6].

Several early findings led to increasing interest in HDAC inhibitors (HDACis) as potent anticancer therapeutics—e.g., histone H4 is commonly deacetylated in human primary malignancies [12] and HDAC family members are frequently overexpressed in a variety of human cancers [13,14,15,16,17].

Of note, the application of HDAC inhibitors proved to be mainly useful in hematological disease [18]. For example, HDAC inhibitor therapy has been shown as an efficacious option in cutaneous T cell lymphoma (CTCL) [19]. Approximately 30% of CTCL patients respond to HDACi treatment, which might be attributable to genetic changes, for example, in adhesion pathways that can also be found in solid tumors [20]. However, outstanding responses in CTCL are more likely due to distinct DNA accessibility features. For example, one observed cluster showed increased access to the HDAC9 locus, responsible for Foxp3-dependent suppression in T regulatory cells. This response to HDACis was highly dependent on transcription factor enrichment of CTCF, affecting overall chromatin structure. In contrast to solid tumors, the unique 3D DNA structure of CTCL rather than oncogenic mutations might be accountable for effective monotherapy with HDACis in this entity [21]. In contrast, monotherapy in solid tumors was found to be largely ineffective, thus directing the focus towards combined inhibition strategies. Nevertheless, HDAC inhibitors still offer great potential as powerful anticancer drugs in solid tumors, and the topic of this review is on current strategies to unlock their potential. The text is also supported by a comprehensive overview of clinical studies involving HDACis (Table A1).

### 1.2. HDAC Subtypes: Structure, Function, Subcellular Localization, and Expression Patterns

The family of histone deacetylases comprises four classes and 18 different isoforms. The single enzymes were numbered according to their discovery, starting with HDAC1 in 1996 [22]. The distinction of the human HDACs into classes is based on their relations to their yeast analogues: class I HDACs show similarities to yeast reduced potassium dependency 3 (Rdp3) protein, class II HDACs are similar to yeast histone deacetylase-A 1 (Hda1), class III enzymes are NAD^+^-dependent histone deacetylases similar to the silent information regulator 2 (Sir2) protein, and class IV only includes HDAC11, with low sequence homology to any of the other HDACs [23]. The HDACs also differ in their subcellular locations and their general expressions, depending on specific tissues. Class I enzymes (HDACs 1, 2, 3 and HDAC8) are localized in the nucleus and expressed ubiquitously [8]. HDACs 4, 5, 7, and 9 (class IIa) show, at least to some extent, cytoplasmatic localization and can be shuttled into the nucleus, whereas HDACs 6 and 10 (class IIb) are primarily localized in the cytoplasm. Members of the class III HDACs can have a broad impact on cellular growth or apoptosis as well; for example, SIRT1 is known to deacetylate, and thus affect, the function the tumor suppressor p53 [24]. The catalytic domains of class I and II HDACs are highly conserved. A similar mechanism of action was proposed for the first time in *Aquifex aeolicus* in cocrystallization studies with HDAC inhibitors [25]. In summary, a water molecule carries out a nucleophilic attack on the carbonyl carbon of the acetylated lysine residue, supported by a polarizing zinc atom and histidine side chain residues. This results in a carbon-nitrogen bond breakage [26].

Furthermore, the activity of histone deacetylases becomes more sophisticated as they exert their activities, usually in huge protein complexes with different biological functions. HDAC1 and -2 act mainly via nucleosome remodeling and deacetylase (NuRD), switch independent 3 (SIN3), mitotic deacetylase (MiDAC) and corepressor of REST (CoREST) complexes, whereas HDAC3 is exclusively recruited by the nuclear receptor corepressor (SMRT/NCoR) complex [27]. The four class I HDACs were reported to act on histones where the vast majority of cellular lysine acetylation takes place [28]. Moreover, even though class IIa HDACs might still play a part in the histone deacetylation process through complex formation with HDAC3, it is now questionable whether they exert any independent deacetylase activity [29]. The class IIb isoform HDAC6 regulates Hsp90, tau and the cytoskeleton through its interactions with tubulin and cortactin, and recognizes ubiquitinated proteins to induce aggresome formation [30,31,32,33,34,35]. HDAC10, the only other class IIb enzyme, functions as a polyamine deacetylase [36]. Since these findings no longer fit into the established classification system, Ho et al. recently suggested the recategorization of HDAC enzymes in accordance with their actual in vitro substrates [29]. Through their versatile and crucial roles in various pathways, HDACs are presumed to contribute to the development of cancer and to other pathological conditions such as neurodegenerative disorders, viral infections and rare diseases [35,37,38,39,40]. There have been plenty of studies demonstrating the aberrant expression of HDACs in different tumor entities. For example, class I HDACs were found overexpressed in prostate [13], renal cell [14], bladder [15] and breast tumors [16]. The latter study also showed HDAC2 and HDAC3 overexpression to be associated with clinicopathological indicators of disease progression. In lung cancer, HDAC3 overexpression was also associated with poor prognosis [17]. In gastric cancer, high class I HDAC expression was related to nodal spread and identified as an independent prognostic marker [41].

### 1.3. Structural Features of Zn^2+^-Dependent HDACs and Development of Subtype-Specific HDACis

So far, crystallographic data available have confirmed a highly conserved nature for the HDAC isoforms. They all feature a variably sized cavity on the surface and a narrow tunnel of mutable length and width, leading to a Zn^2+^ ion located in the active site [36,42,43,44,45,46]. In accordance with the structural characteristics of the different isoforms, a reliable pharmacophore for HDAC inhibitors has been established [47]. As illustrated using the example of vorinostat (Figure 1), HDAC inhibitors (HDACis) typically comprise a cap group occupying the entrance area to the active site, a zinc binding group (ZBG) interacting with the Zn^2+^ ion in the catalytic center and a hydrophobic linker connecting the two units through the enzyme’s channel rim [48]. This structural motif can be generally applied to all isoforms, and when structural information on the different isoforms was rare, most design attempts were based on this pharmacophore. Thus, inhibitors of the first generation generally turned out to be rather unselective. Meanwhile, several structural characteristics have been found that can be specifically addressed for discriminating between the different isoforms. A 14 Å long cavity adjacent to the zinc ion, the so-called “foot pocket”, has only been observed in class I HDACs, allowing suitable 2-aminoanilide ZBGs to target them [43,49,50,51]. HDAC8, on the other hand, is smaller than HDACs 1–3 and lacks a C-terminal domain, which is required for joining multiprotein complexes [29]. A characteristic mutation present in the active sites of all class IIa isoforms is the replacement of a catalytically crucial tyrosine by a histidine residue which can rotate to open the lower pocket [52,53]. A crystal structure of the class IIa isoform HDAC7 further revealed the existence of a hydrophobic binding site in proximity to the active site, which might be required for protein–protein interactions [54].

HDAC6 is the only isoform possessing two functional catalytic domains and an additional zinc finger, serving as an ubiquitin-binding domain (HDAC6 UBD) [55]. For the second catalytic domain (CD2) of HDAC6, the tunnel appears to be slightly more spacious but shorter, while the entrance area on the enzyme’s surface is larger compared to other isoforms [46]. Selectivity for HDAC6 can therefore be achieved by incorporating large cap groups and benzyl linkers instead of aliphatic chains.

With the information obtained by X-ray cocrystal structure analysis and comprehensive SAR studies, remarkable progress in the field of isoform-selective compounds has been made over recent years [48,56]. Class I-selective HDACis such as entinostat were the first group of selective HDACis to enter clinical trials and are now widely investigated for the treatment of cancer and nononcological diseases. In 2015, the 2-aminoanilide tucidinostat was the first class I-selective compound to be approved by the NMPA in China and is now undergoing late-stage studies in Europe and the US. Further Phase I and Phase II studies investigate the effects of the HDAC6-preferential candidates ricolinostat and citarinostat on multiple myeloma and other malignancies, including solid cancers.

## 2. Molecular Mechanisms of HDACi-Promoted Anticancer Effects

### 2.1. Apoptosis Induction

Induction of apoptotic cell death represents one of the main mechanisms of anticancer effects promoted by HDAC inhibitors [57,58,59,60,61]. Regarding apoptosis signaling in cancer cells upon treatment with HDAC inhibitors, divergent effects have been demonstrated depending on the cellular context. This can result in desirable proapoptotic or in unwanted antiapoptotic responses. This ambiguous impact on cancer cells with respect to apoptosis parallels the effects of HDAC inhibitors regarding the promotion of autophagy or senescence, which may also result in tumor inhibiting or tumor promoting effects. In the best-case scenario, treatment with HDAC inhibitors induces growth arrest or cell death of cancer cells; however, under unfavorable conditions, these agents can impede the effect of other anticancer therapeutics or may even promote tumor growth.

The complexity of HDAC inhibitor-dependent responses with regard to apoptosis arises from the fact that these compounds affect virtually all components of the apoptotic machinery. For example, HDAC inhibitors lead to the upregulation or sensitization towards factors of the extrinsic apoptosis pathways [62,63,64,65]. Moreover, HDAC inhibitors can increase the expression or activity of proapoptotic proteins of the intrinsic pathway, such as Bax or Bak [63,66], or decrease the expression of antiapoptotic Bcl proteins such as Bcl-2 or Bcl-xL [67,68].

One important mechanism of how HDAC inhibitors affect apoptotic signaling is the generation of reactive oxygen species (ROS) [69,70]. The increase in oxidative stress upon HDAC inhibition can result from the upregulation of ROS-producing pathways or from the downregulation of endogenous antioxidant factors such as thioredoxin [71,72,73]. The importance of ROS generation for HDAC inhibitor-promoted cell death is underlined by the observation that ROS scavengers such as N-acetyl cysteine (NAC) could decrease the cytotoxicity of HDAC inhibitors in various cancer cell lines [74,75,76]. However, it has to be noted that vitamin C or NAC treatment did not affect apoptosis induction in HeLa cells treated with suberoyl bishydroxamic acid [77]. Some discrepancies in the relation of ROS generation and apoptosis upon HDAC inhibition may be related to the fact that oxygen stress can play a causative role in apoptosis induction but can also be a consequence of ROS-independent apoptosis [78]. Moreover, ROS-dependent and -independent effects of HDAC inhibitors can lead to the adaptive stimulation of the Keap1-Nrf2 pathway, which is a central regulator of antioxidant molecules [79,80,81,82]. This effect has been mainly described in nononcological conditions, but may be also relevant in cancer. Since Nrf2 is a transcription factor responsible for the upregulation of cytoprotective factors [83,84,85], the activation of the Nrf2 pathway by HDAC inhibitors could even result in unwanted prosurvival effects in cancer cells [86].

Another pivotal effector in this context, which is critically affected by HDAC inhibitors, is p53 [87]. Treatment of cancer cells with HDAC inhibitors can increase p53 expression and restore p53 activity [88,89,90] mainly by downregulation of MDM2 or MDM4 [91,92] or by increasing p53 acetylation [92]. The latter effect is an excellent example of histone-independent effects of HDAC inhibitors. Acetylation of p53 is a post-translational modification crucial for p53 function [93] and HDAC1 is a central regulator of this modification [94]. It appears plausible that restoration of p53 by HDAC inhibitors is particularly effective in tumor cells with wild-type p53 status, whereas this mechanism is without relevance in tumor cells with genomic p53 deletion and may even be detrimental in tumor cells with p53 mutations. However, the anticancer effects of HDAC inhibitors cannot be simply deduced from the p53 status of the cancer cells. For example, the pan HDAC inhibitor vorinostat increased the radiosensitivity of wildtype (wt) p53 glioblastoma cells, but this effect was not seen in p53-mutated cells [95]. In contrast, vorinostat increased the expression of p53, but was equally effective against wt p53 and p53-deleted colon cancer cells, whereas the efficacy of class-I-selective inhibitor entinostat was dependent on p53 status [91]. Of note, vorinostat led to a downregulation of mutated p53 in colorectal cancer cells, but to an upregulation of wt p53 [96]. The notion that effects on p53 differ with cellular context is emphasized by the fact that in MDM2-amplified liposarcoma cells, HDAC inhibition resulted in the downregulation of p53 irrespective of its mutational status, whereas in cancer cells without MDM2 alteration no effect on p53 expression was observed [97].

### 2.2. Autophagy Induction

Several studies have investigated the role of autophagy in response to the inhibition of histone deacetylases. Yet, it is still not clear whether increased induction of autophagy promotes cell death or serves as a survival mechanism upon cellular stress. To some extent, the effects of HDACis on cell viability have been shown to be nonapoptotic since knockdown of proteins required for apoptosis, such as Apaf-1, or the inhibition of caspase via Z-VAD-FMK was not able to prevent vorinostat-induced cell death [98]. Unifying several studies, an increase in autophagic flux has been observed, as monitored by alterations in the levels of proteins defined as hallmarks of autophagy. This includes increases in LC3-II or Beclin-1 and a decrease in p62 [99,100,101,102,103,104,105]. In Tamoxifen-resistant breast cancer cells, vorinostat treatment led to cell death with little activation of apoptosis, while elevated levels of LC3-II and Beclin-1 were seen. Inhibiting the autophagic flux with 3-Methyladenine (3-MA) led to significantly enhanced cytotoxic potential of vorinostat [105]. It remains unclear whether an intact estrogen receptor is necessary to mediate these effects [100], since in triple negative breast cancer (TNBC) cells, cotreatment with the autophagy inhibitor chloroquine and panobinostat led to prolonged survival of tumor bearing mice as well [106]. Autophagy induction has been shown to be mediated via the transcription factor FOXO1. FOXO1 knockdown prevented the expression of autophagy-related genes upon HDAC inhibition and the repeated inhibition of autophagy generated led to enhanced cell death [99]. Glioblastoma stem cell xenografts were very sensitive towards HDAC inhibition with a more than 35% reduction in tumor size. While vorinostat caused autophagy induction, the knockdown of beclin-1, LC3 or ATG5 revealed again a dramatic increase in vorinostat-mediated apoptosis [103]. In summary, activation of autophagy upon HDAC inhibition is often seen as survival mechanism which needs to be abolished for continued apoptotic or nonapoptotic cell death [107].

In contrast, the broad-spectrum inhibitor panobinostat induced autophagy and cell death in hepatocellular carcinoma (HCC). Here, its efficacy was further increased by tamoxifen, an autophagy inducer, rather than by the inhibitors 3-MA and bafilomycin, which could even reverse cytotoxic effects of HDACis [104,108]. In difficult to treat entities such as chondrosarcoma, vorinostat showed a growth inhibitory effect. For this, intact autophagic flux was required since 3-MA application reversed the vorinostat-induced cell death [109].

Resistance to chemotherapy is an often-experienced clinical problem and the contribution of autophagic signaling might be involved in this observation. In nonsmall cell lung cancer (NSCLC) cells, HDACis increased the cytotoxicity of pemetrexed. Interestingly, simultaneous treatment led to antagonistic effects while sequential therapy with pemetrexed followed by HDACis was the most effective. It also showed a morphologic increase in acidic vesicular organelles, a typical marker of autophagy [110]. In lung cancer patients, IGFB2 has been identified as predictive for chemoresistance. Trichostatin A (TSA) decreased IGFB2 expression in lung cancer cells and resensitized cells towards cisplatin treatment. TSA also led to increased LC3 expression and the degradation of p62, enhancing autophagy flux [111]. Recently, entinostat in combination with cisplatin showed synergistic effects in esophageal squamous cell carcinoma. In this case, the addition of the autophagy inhibitor 3-MA increased cell death, again indicating the protective properties of increased autophagy [102]. In line with previous context-dependent effects, vorinostat was able to degrade mutant p53 in some cancer cells, albeit others did not show this phenomenon. In TNBC, autophagy correlated with mutant p53 degradation; however, this observation could not explain vorinostat sensitivity when compared to other cells [101].

### 2.3. Senescence Induction

Cellular senescence is a stable cell cycle arrest. It can be described as a phenotype provoked by different stressors—for example, oncogene-induced senescence, therapy-induced senescence or telomere shortening. Senescent cells can be resistant to apoptosis, remain metabolically active and secrete various factors for communication with the tumor microenvironment [112]. A characteristic marker to identify these cells is an increased activity of the senescence-associated β-galactosidase.

Different HDACis have been found to induce senescence; however, the clinical implication is not yet fully understood. In rhabdomyosarcoma, vorinostat was shown to mediate senescent features and increase the expression of the cell cycle inhibitors p21 and p27 [113]. Urothelial cancer cells revealed senescence morphologies upon treatment with unselective HDACis [114]. In HCC, senescence induction was dependent on the inhibitor used, with vorinostat or valproic acid (VPA) [115] being able to increase SA-β-galactosidase activity while TSA was not [116]. In adenoid cystic adenocarcinoma, the combination of vorinostat and cisplatin was highly effective and revealed characteristic senescence features [117]. A study in breast cancer described miR-31 as a target of HDACis and a highly relevant regulator of cellular senescence [118]. In glioblastoma stem cells, low concentrations of vorinostat led to cell cycle arrest without affecting apoptotic markers to a larger extent. p38 activation and subsequent p53 phosphorylation seemed to be involved in this observation [119]. In head and neck squamous cell carcinoma (HNSCC) and nonsmall cell lung cancer (NSCLC), some cells became senescent in response to cisplatin or taxol treatment and showed elevated BCL-xL expressions. In this case, panobinostat was used as a senolytic drug and, interestingly, was found to be more efficient at killing cancer cells than a second cycle of the previously administered chemotherapy [120]. In conclusion, it is clear that senescence has to be taken into account when exploring HDACi-induced effects, albeit it is still not well-understood.

### 2.4. Effects on DNA Damage

The impact of HDAC inhibitors on DNA integrity is another pivotal aspect with regard to cytotoxic responses upon treatment with these agents [37,121]. In some cases, the appearance of DNA lesions after exposure to HDAC inhibitors may be merely a consequence of their proapoptotic or cytotoxic effects. However, DNA damage may often represent an initial and critical molecular event that is directly responsible for the anticancer effects of HDAC inhibitors.

A better understanding of the still not fully resolved mechanisms behind the occurrence of DNA damage after HDAC inhibitor treatment is a central issue with regard to the therapeutic use of these substances and the development of novel inhibitors. In fact, HDAC inhibitors have been described as genotoxic or mutagenic agents in a number of reports in malignant [122,123,124] as well as in nonmalignant cells [125,126,127,128]. From these findings, the question arises as to whether HDAC inhibitors have a carcinogenic potential, which would be especially relevant when considering their therapeutic use in younger patients and/or in nononcological diseases. Although the DNA damaging effects of HDAC inhibitors have been found to be more pronounced in malignant than in nonmalignant cells according to some reports [129,130], this issue clearly demands further clarification.

Regarding the mechanisms of DNA damage induced by HDAC inhibitors, two plausible explanations have been proposed: (1) the induction of oxidative stress by HDAC inhibitors and (2) the inhibition of the DNA repair machinery, with the subsequent accumulation of DNA lesions evoked by endogenous or exogenous mutagens.

The issue of oxidative stress induction by HDAC inhibitors has already been discussed above in the context of apoptosis induction (see 2.1). An involvement of reactive oxygen or nitrogen species in DNA damage related to HDAC inhibitors is suggested by the fact that an increase in markers of oxidative DNA stress has been associated with HDAC inhibitor treatment [131,132]. One important question to be addressed in this context is whether oxidative stress elicited by HDAC inhibitors is a consequence of the enzyme inhibition by these agents or a result of reactive decomposition products of the inhibitor molecule itself. In the latter case, at least some genotoxic effects of HDAC inhibitors would be independent of HDACs and show significant differences, depending on the chemical structure of the inhibitor molecule. Of note, hydroxamic acid derivatives (also representing one important class of HDAC inhibitors) may under certain conditions give rise to isocyanates [133,134,135], which can directly or indirectly lead to DNA modifications. Moreover, hydroxamic acids may release nitrogen monoxide [136], which could be also responsible for an increase in oxidative stress. However, it is still unsettled as to whether these reactions also occur under physiological conditions in a living cell.

The second pathway, which has been thought to result in an accumulation of DNA damage, is the interference with the DNA repair mechanism by HDAC inhibitors [137,138,139,140]. This effect of HDAC inhibitors suggests a synergism upon combination with DNA damaging chemotherapeutics, providing a rationale for the generation of hybrid molecules with combined HDAC inhibiting and DNA alkylating properties, as discussed below (see 4.3). From a mechanistic point of view, impairment of DNA repair by HDAC inhibitors may be a consequence of (1) altered DNA architecture or (2) dysregulated expression or activity of DNA repair enzymes and/or DNA damage signaling.

With respect to the structural organization of DNA, histone modifications are a key element affecting the DNA access of mutagenic substances, but also of repair proteins. Thus, chromatin organization is of paramount importance for the integrity of the genome [141,142,143]. The next level, i.e., the HDAC-dependent regulation of the expression and function of the DNA repair machinery, is a highly complex issue, since virtually all types of DNA repair mechanisms are impacted by HDACs [140,144]. Thus, the in-depth discussion of DNA repair proteins that have been shown to be affected by HDAC inhibitors would go far beyond the scope of this review. However, two important aspects should be mentioned in brief.

Firstly, HDAC1 has been shown to directly stimulate oxoguanine glycosylase 1 (OGG1), a repair protein critically involved in base excision of oxidized guanine residues, whereas HDAC1 deficiency causes impairment of OGG1 activity [145]. Thus, an increase in 8-oxoguanine in DNA after HDAC inhibition could be the consequence of increased oxidative stress (see above) or impaired repair of this lesion. Therefore, the detection of 8-oxoguanine lesions after HDAC inhibitor treatment may be insufficient for proving that this agent augments oxidative stress per se.

Secondly, the repair of one of the most lethal DNA damages, the induction of DNA double strand breaks, is critically regulated by HDAC subtypes 1 and 2, which are directly recruited to DNA damage sites, whereas other HDAC isotypes such as HDAC3 are not involved in this process [146]. From this finding, HDAC1/2 subtype-specific inhibitors should be exceptionally well-suited for combination therapies with DNA damaging agents inducing double strand breaks.

### 2.5. Effects on Hormone Signalling

In hormone-dependent cancers such as breast or prostate cancer, early evidence suggested that HDAC inhibition affects the expression of hormone receptors such as the estrogen receptor (ERα, ERβ) and the progesterone receptor (PR), as well as the androgen receptor (AR). Broad-spectrum HDAC inhibitors are able to reduce ERα expression in hormone receptor expressing cells [147,148,149,150,151,152]. Consequently, the expression of ERα-induced genes via stimulation with estradiol could be antagonized by up to 88% upon treatment with valproate or TSA [147]. Proliferation stimulating effects provoked by estradiol, e.g., upregulation of cyclin D or hyperphosphorylation of pRB, were efficiently abrogated upon vorinostat treatment [148]. ER+ cells appear to be more sensitive towards proliferation inhibition than ER- cells. This might be attributed to stronger p21 expression upon HDAC inhibition in ER+ cells [149]. Under hypoxic conditions, more pronounced ERα downregulation under HDACi treatment has been observed. This response was highly dependent on an intact proteasome pathway [152]. In the course of broad-spectrum HDAC inhibition, HDAC6 function is blocked as well. This results in Hsp90 hyperacetylation with further ERα destabilization, leading to higher levels of polyubiquitinylated hormone receptors and subsequent proteasomal degradation. ERα degradation via HDACis can be reversed by proteasome inhibitors such as bortezomib [153]. This mechanism still needs further investigation since ERα downregulation was also detected upon specific HDAC6 inhibition, regardless of HDAC6 enzymatic activity [154].

Additionally, it has been shown that TSA and raloxifene can induce a strong ERβ upregulation in Erα-positive cells while simultaneously reducing ERα expression. This may also account for the better growth inhibitory effects of HDAC inhibitors since exogenous ERβ transfection increased cell susceptibility towards TSA-mediated growth inhibition [155]. While ER+ cells typically respond to hormone treatment, TNBC does not and so behaves more aggressively. Numerous studies have shown that ER expression can be restored via HDACis [156,157,158,159,160]. These effects can differ depending on the inhibitor used. The broad-spectrum inhibitor TSA increased estrogen response element activity in TNBC cells, with PR upregulation induced by increased ERβ expression. This may contribute to their higher susceptibility towards tamoxifen [157]. The class I-selective inhibitor entinostat increased ERα and aromatase protein expression in TNBC cells. Upon further stimulation with estradiol, the expressions of estrogen receptor target genes (e.g., pS2) were significantly enhanced and could be blocked by letrozole, resulting in highly reduced tumor growth in vivo [156]. Furthermore, it may not be required to use a broad-spectrum HDACis for inducing ER expression in ER- cells. Indeed, the knockdowns of HDAC1, HDAC2 and HDAC3, alone or in combination, mediated efficient ER upregulation in some TNBC cells, and HDAC2 expression was found to be negatively correlated with ER expression [161].

On the other hand, studies in TNBC patient-derived xenografts treated with different HDAC inhibitors failed to demonstrate a marked increase in ERα nor ERβ expressions, thus not supporting a role for combined HDAC inhibition and hormone therapy [162]. In endometrial cancer cells, albeit lacking ERα, the predominant PR inducer panobinostat was able to reintroduce PR expression, which also repressed the oncogene myc. Panobinostat could thus also lead to restored sensitivity towards hormone therapy [163]. On the mechanistic side, an increased cell cycle arrest in the G1 phase has also been observed upon HDAC inhibition in endometrial cancer cells when combined with progesterone [164].

Hormone receptors play a pivotal role in prostate cancer as well. HDAC inhibitors have been shown to suppress the expression of AR target genes. Although HDACis can reduce AR expression by themselves [165], the main mechanism for the decrease in AR-induced gene expression was described as blockage of RNA polymerase II recruitment to HDAC-dependent promotors [166]. AR activity can also be repressed via post-transcriptional regulation, since miR-320a was found to reduce AR levels and to inhibit growth. The HDACi OBP-801 was identified as being directly involved in miR-320a upregulation in prostate cancer cells [167]. Belinostat appeared to be more effective in androgen-sensitive tumors than in androgen-independent ones. The introduction of wild-type AR into AR- cells enhanced the cytotoxic effects of belinostat [168]. In up to 50% of all prostate cancers, a gene fusion between the androgen responsive TMPRSS2 gene and the oncogenic transcription factor ERG can be observed. The suppression of this fusion gene by androgen deprivation was further enhanced by HDAC inhibition and resulted in synergistic growth inhibition in prostate cancer cells [169].

### 2.6. Immune Effects

Nowadays, immunotherapy plays an eminent and ever-growing role in the treatment of cancer patients. HDAC inhibitors can influence the immune system in many ways, often depending on the cellular context and the tumor microenvironment. It has been shown that HDACis can affect dendritic cell activation and the antigen presenting machinery, as well as T-cell activation or the presence of regulatory T-cells and myeloid-derived suppressor cells in the tumor microenvironment. Thus, they may have an impact on the efficacy of immune checkpoint inhibitor treatment [170]. A screening in lung adenocarcinoma for substances inducing T-cell chemokine expression identified HDACis as capable of doing so. Romidepsin or vorinostat induced mRNA levels of chemokines Ccl5, Cxcl9 and Cxcl10 in KRAS-mutant cells. These findings were confirmed in xenografts which showed increased T-cell infiltration upon romidepsin treatment. The antitumor effect was also dependent on T-cells, since anti-CD4 and anti-CD8 antibodies reversed the antiproliferative effects. Romidepsin and anti-PD1 treatments proved to be synergistic and to be dependent on IFN-γ [171].

The necessity of T-cell infiltration or activation for mediating antitumor response was shown in different studies. In an orthotopic bladder cancer model, intravesical HDACi instillation and systemic PD-1 blockage led to curative responses. CD8 depletion fully abrogated the growth inhibitory effects [172]. Regulatory T-cells (Tregs) and myeloid-derived suppressor cells (MDSCs) are known to suppress immune responses. In models of breast or pancreas cancer, the combination of entinostat and anti-PD1 treatment significantly prolonged survival. The treatment reduced the number of granolytic MDSCs and their ability to negatively affect T-cell proliferation by producing less Arg-1. Entinostat and immune-checkpoint inhibition (ICI) therapy modulated various pathways in G-MDSCs, including mTOR, ERBB and VEGF signaling, eventually resulting in increased infiltration of granzyme-B-producing cytotoxic T-cells [173]. The class I-selective HDACi domatinostat increased the levels of antigen-processing machinery (APM) genes and upregulated MHC-I and -II and IFN-γ response genes in melanoma cells. PD-1 blockade was enhanced upon HDAC inhibition. In melanoma patients, biopsies taken after 14 days of domatinostat treatment showed an enhanced pembrolizumab response signature in 4 out of 6 patients compared to biopsies before treatment. Upregulation, for example, of IFN-γ was similar to the murine in vivo findings, thus strengthening further clinical trial application [174].

Additionally, Treg depletion is considered as responsible for improved ICI results upon HDACi treatment [175]. TNBCs were cocultured with peripheral blood mononuclear cells. In the coculture, the amount of Foxp3-positive Tregs was higher than in peripheral blood mononuclear cells (PBMCs) cultured alone. Vorinostat treatment mitigated this effect. Interestingly, a TNBC model highly resistant to ICI showed antitumor response when ICI (using anti-PD1 and anti-CTLA4) was augmented by HDACi treatment [176]. Decreased Tregs and MDSCs were also observed in PBMCs of melanoma patients [177] or in renal and prostate cancer models. The latter showed Treg suppression with low-dose entinostat, while not affecting T effector cell proliferation [178]. Although it is obvious that T-cell activation exerts an important role in HDACi-induced immune response, other studies also highlight the activation of natural killer (NK) cells. Entinostat rendered prostate cancer cells more susceptible to avelumab-mediated antibody-dependent cellular cytotoxicity when the cells were treated with NK cells from heathy donors. In addition, NK activating ligands were upregulated and NK cells from cancer patients shifted towards more active states [179].

Various studies have been published on PD-L1 or PD-L2 expression upon HDACi therapy, with different results. While in ovarian PDX models HDAC knockdown of mainly class I members reduced PD-L1 and PD-L2 expressions [180], HDAC6 inhibition or knockdown of HDAC6 in colorectal cancer cells decreased IFN-γ-induced PD-L1 expression [181]. In another study, the HDAC6 inhibitor nexturastat A reduced PD-L1 expression provoked by anti-PD1 treatment and subsequently depleted protumorigenic M2 macrophages. In combination with PD1 blockade, it increased the total number of tumor infiltrating CD8+ lymphocytes [182]. In contrast, in primary melanoma cells, class I-selective HDACis dose-dependently increased PD-L1 expression. Melanoma cell lines showed elevated histone acetylation at the PD-L1 promoter upon panobinostat treatment [183]. Taken together, several different mechanisms regarding the impact of HDACis on the immune system have been proposed. While appearing to be context-dependent, they clearly indicate a role for HDACis in improving ICI results.

## 3. Combination Strategies

### 3.1. Combination with mTOR Inhibitors

With regard to HDAC- and mTOR inhibitor combination therapies, preclinical data often suggest synergistic effects. However, only a small number of clinical trials have been published so far. In the rare tumor entity synovial sarcoma, the mTOR inhibitor ridaforolimus led to increased p-AKT levels. This was abrogated by cotreatment with vorinostat and resulted in synergistic cytotoxicity [184]. In renal cell carcinoma, vorinostat enhanced the activity of temsirolimus. This was mainly attributed to a decrease in survivin, leading to a more pronounced induction of apoptosis, and the reduction in VEGF in two xenograft models [185]. Histological studies in prostate cancer and NSCLC also showed reductions in the proliferation marker Ki67 upon combination treatment, next to reduced migration, adhesion and invasion capacity [186,187,188].

Moreover, combination effects of mTOR inhibitors and HDAC inhibition were seen independent of the AR status, thus potentially offering an opportunity for hard-to-treat, castration-resistant prostate cancer [189]. In temsirolimus-resistant cells, valproate reduced the migration potential. An underlying alteration of integrin α5 expression could be responsible for this observation [190]. Valproate also counteracted the resistance of bladder cancer cells to temsirolimus. In this case, a reduction in cell cycle regulators such as cyclin A and CDK2 was deemed responsible [191]. Furthermore, the combination of trichostatin A and an mTORC1/2 inhibitor (MLN0128) inhibited the growth of malignant breast cancer cells more efficiently compared to nontumorous mammary epithelial cells [192]. In a patient-derived xenograft model and in patient tumor slices of TNBC, the triple combination comprising valproate, tamoxifen and rapamycin was capable of inhibiting growth as well [193].

Interestingly, the combination of vorinostat and the mTOR inhibitor sapanisertib was efficacious in NF-1 mutant nervous system malignancies. This was repeatedly confirmed by using different HDAC and mTOR inhibitors. On the mechanistic side, a “catastrophic” oxidative stress response was observed. The mRNA of thioredoxin interacting protein (TXINP), which inhibits the important antioxidant thioredoxin, was drastically upregulated. This combination also led to tumor regression in KRAS mutant NSCLC xenografts [194]. In a Phase I trial, vorinostat and sirolimus were combined for the treatment of advanced malignancies. One patient with perivascular epitheloid tumor presented with a 54% reduction in target lesions after six cycles. Two other patients with fibromyxoid sarcoma and HCC experienced stable disease for over 12 months [195]. In another Phase I trial, combined panobinostat and everolimus treatment was evaluated in clear cell renal cell carcinoma. In total, 21 patients were enrolled. There was no objective response but stable disease in 13 patients. Four patients with stable disease had to discontinue treatment because of toxicity. The study also found a higher baseline expression of miR-605 in patients with progressive disease compared to patients with stable disease [196]. However, a rationale for combined therapy may still exist since some patients experienced surprisingly good responses. The combination of vorinostat and ridaforolimus yielded in stable disease in 4 out of 15 patients, lasting up to 80 months. Noteworthily, three of these patients had progressed on prior mTOR inhibitor therapy [197]. The main problem is to identify these patients responding in this niche setting. Dual targeting inhibitors have been synthesized to facilitate the combination of these drugs [198].

### 3.2. Combination with Kinase Inhibitors (EGFR, PI3K)

Targeted therapies provide reasonable therapeutic opportunities for patients with alterations or aberrant expressions of growth factor receptors or downstream signaling pathways. Several studies implicate a benefit from adding HDACis to EGFR inhibition, even in resistant cells. Convincing preclinical data provided the rationale for clinical trials combining these drugs.

In NSCLC, gefinitib plus vorinostat induced cell death synergistically in EGFR exon 19- or T790M-mutated cells. Notably, this was also true for gefitinib-resistant cells. A downregulation of HSP90 was observed, followed by a decrease in EGFR and even in MET protein expression. This was dependent on an increase in ROS-driven caspase activity [199]. Sometimes, BIM deletion polymorphism can be responsible for EGFR TKI resistance in NSCLC. Tanimoto et al. showed that vorinostat affects the alternative splicing of BIM mRNA in the deletion allele. The combined use of osmertinib and vorinostat reintroduced BIM expression and led to apoptosis. This was also observed upon HDAC3 knockdown and further confirmed by a marked tumor growth inhibition in BIM-deleted xenografts [200].

KRAS mutant lung cancer shows an aggressive phenotype. The drugs tucidinostat and icotinib, approved in China, reduced cellular growth of both, wild-type and KRAS mutant NSCLC cells, while slightly reducing p-Akt and p-MAPK levels [201]. Panobinostat showed the same potency in overcoming gefitinib resistance in KRAS-mutated cells. These effects were attributed to reduced levels of TAZ, an important protein in the hippo pathway, upon HDACi application [202]. The combination of panobinostat and another first generation TKI, erlotinib, showed similar results, sensitizing cells towards the TKI treatment. In some cells, a shift towards an epithelial phenotype was observed as well, with elevated E-cadherin levels upon HDAC inhibition [89].

In 2010, the multitarget inhibitor CUDC-101 was designed, inhibiting not only HDAC but also EGFR and HER2 [203]. Activity was proven in 54 cell lines, and additionally suppressed compensatory activation of HER3 or MET by reducing protein levels [204]. In anaplastic thyroid cancer, a screen of >3000 drugs identified CUDC-101 as potent. It induced cell death, attenuated the MAPK-signaling pathway, upregulated p21 and further reduced levels of antiapoptotic proteins—e.g., survivin or XIAP [205]. The latter finding was also observed in pancreatic cancer where CUDC-101 proved to be an even a better radiosensitizer than vorinostat [206]. Moreover, the combined inhibition of HDAC, EGFR and HER2 intensified cytotoxicity of gemcitabine, an obligatory drug in pancreatic cancer. EMT was reversed as E-cadherin levels were increased, while mesenchymal markers such as Vimentin and MMP-9 and the suppressive transcription factors Snail and Slug were decreased [207]. The reversal of EMT was also noted in another study where vorinostat sensitivity was in line with the induced decline of p63 transcription factor and EGFR protein level [208]. In hard-to-treat glioblastoma, erlotinib sensitivity was restored by HDACis and CUDC-101 was found to lead to lower EGFR mRNA and protein levels. Interestingly, no acquired resistance was observed [209].

Beyond growth factor receptors, drugs for the inhibition of distant signaling molecules or kinases in combination with HDACis have been described as well. The compound CUDC-907 (fimepinostat) merges phosphoinositide 3-kinase (PI3K) inhibition with HDACis [210]. For example, proliferation of chronic lymphatic leukemia cells was inhibited via downregulation of antiapoptotic Bcl-2 family members [211]. CUDC-907 evades drug resistance and also showed effects in myc-driven lymphoma cells [212]. Next to the increase in apoptosis in prostate cancer cells, CUDC-907 also mediated antiproliferative effects through inducing DNA damage [213]. Multiple entities have shown sensitivity towards fimepinostat, thus it is further evaluated in Phase I/II clinical trials [214]. In pancreatic cancer cells, CUDC-907 reduced tumor growth in vitro and in vivo. This study demonstrated HDAC6 inhibition with subsequent degradation of the oncogenic transcription factor c-myc. The cytotoxic potential was impaired by proteasome inhibition via MG132 [215]. CUDC-907 revealed synergistic trends also in other entities, by reducing tumor growth of prostate cancer patient-derived xenografts up to 60%, again decreasing c-myc levels and evoking significant DNA damage. It also repressed thyroid cancer cell growth [216] and synergized with olaparib in SCLC where DNA double strand breaks were found to accumulate [217]. Additionally, it reversed cisplatin resistance by downregulating multidrug resistance protein ABCC2 [218]. A broad spectrum of studies, as above, have paved the way for clinical trial evaluation.

In a first-in-human study, the above-described drug, CUDC-101, was well-tolerated. Of 25 enrolled patients with different tumors, 15 were available for efficacy. One patient with metastatic gastric cancer had a partial response with a 56% decrease in the target lesion, and two other patients with head and neck cancer showed a >20% decrease in tumor volume in the target lesions. Six patients in total experienced stable disease as the best response [219]. A Phase I trial with 12 patients with locally advanced HNSCC combined CUDC-101 with chemoradiation. In total, 90% of the patients did not show disease progression until a median of 1.47 years [220].

In heavily pretreated metastatic HER2-positive breast cancer, entinostat could be safely administered in combination with lapatinib and (optionally) trastuzumab. Out of 35 patients, complete response was achieved in three patients, partial response in three patients and one patient had stable disease [221]. In another study, vorinostat and trastuzumab showed safety but inefficient drug activity [222]. Out of 42 patients with aerodigestive tract tumors, 33 were evaluable for efficacy in a Phase I trial of panobinostat and erlotinib administration. The most common adverse events were fatigue, nausea and rash. The disease control rate of 54% consisted of three patients with partial responses and 14 with stable disease. EGFR mutant patients showed the best overall survival. Seven patients with prior TKI therapy had been reviewed; five were evaluable for efficacy, including three patients with slow progression over at least six cycles and two with rapid progression [223]. Fifty-two pretreated patients with advanced NSCLC were recruited for a Phase I/II trial on gefitinib and vorinostat; 16 partial responses and 6 patients with stable disease were noted. This provided the rationale for the ongoing investigation of this drug combination in lung cancer [224].

Another big Phase II trial tried to elucidate the role of entinostat in combination with erlotinib in patients who progressed on prior chemotherapy. The 4-month progression-free survival rate showed no significant difference in the entinostat vs. placebo group (18% vs. 20%). Patients with E-Cadherin IHC3+ staining, however, experienced an overall survival of 9.4 months with entinostat compared to 5.4 months with placebo [225]. In summary, the combination of HDACis with EGFR or PI3K inhibition shows promising preclinical activity and some evidence for a clinical benefit. Yet, a reliable biomarker to better stratify patients likely to respond is still missing.

### 3.3. Combination with Selective Estrogen Receptor Modulators (SERMs) and Antiestrogens

A decent number of studies have investigated the effects of HDAC inhibitors on breast cancer cells, showing a definite influence on the hormone receptor expression and consequently paving the way for a combined inhibition with superior antiproliferative effect. Furthermore, HDACis were able to aid in turning aggressive TNBC into hormone responsive tumors. In ER- breast cancer xenografts, combined treatment with entinostat and letrozole exerted the most pronounced effect compared to single compound treatment. Mice treated with entinostat showed increased ER and aromatase expressions, probably sensitizing cells towards letrozole [156,226,227]. A study in ER+ cells showed enhanced antiproliferative potential when trichostatin A or vorinostat were combined with tamoxifen [228].

HDACis may also be efficacious when the common problem of acquired resistance inevitably arises throughout the course of treatment. Panobinostat was able to block proliferation of aromatase inhibitor-resistant breast cancer cells. The effects were mainly attributed to the downregulation of NF-κB, which was shown to be permanently active in the above-mentioned resistant cells [229]. There have also been efforts to develop single molecules which show combined estrogen- and HDAC inhibition. These compounds are often associated with higher activity and specificity than the single drugs. In fact, the hybrids characteristically showed greater cell type selectivity, leaving noncancerous cells unaffected [230,231,232].

In a Phase I trial, metastatic breast cancer patients were treated with letrozole and panobinostat. Out of 12 patients, five showed stable disease and two a partial response. Interestingly, these patients had progressed on prior endocrine or chemotherapy. Even one patient with triple negative breast cancer showed disease stabilization [233]. One hundred and thirty patients with locally recurrent or metastatic estrogen receptor-positive breast cancer participated in a Phase II trial, either receiving exemestane plus placebo or exemestane plus entinostat. The median overall survival was 28.1 months in the entinostat group vs. 19.8 months in the placebo group (hazard ratio (HR) 0.59; 95% CI 0.36 to 0.97; *p* = 0.036). The most common adverse events included fatigue, nausea and neutropenia. Serious adverse event incidence was similar in both groups. Protein hyperacetylation was associated with longer progression-free survival (PFS) and might serve as a biomarker for future trials [234].

Another study investigated the combination of vorinostat and tamoxifen in breast cancer patients with resistance to hormone therapy. Forty-three patients who had received prior chemotherapy, tamoxifen or aromatase inhibitors were enrolled. The combination proved safe and the overall response rate (ORR) was 19%, while another 21% showed stable disease. Two patients with disease stabilization experienced complete metabolic responses, as detected by FDG-PET. Histone H4 acetylation was associated with higher response rates next to higher HDAC2 expression in responding vs. nonresponding patients [235]. To date, one Phase III trial has been published which investigated the role of tucidinostat, an oral subtype-specific inhibitor preferentially targeting HDACs 1, 2 and 3. The trial included hormone receptor-positive and HER2-negative patients with advanced breast cancer who progressed after at least one endocrine therapy. In total, 365 patients were enrolled, receiving either exemestane plus placebo or exemestane plus tucidinostat. Independent review imaging assessment revealed a median progression-free survival of 9.2 months in the tucidinostat group vs. 3.8 months in the placebo group (HR 0.71; 95% CI 0.53–0.96, *p* = 0.024). Most common adverse events in the tucidinostat group were hematological effects such as neutropenia or thrombocytopenia. Overall, this study established tudicinostat as a new treatment option for hormone therapy refractory patients with the limitation of only including one ethnical group [236]. Notably, the recent NMPA-approval in China of tucidinostat in combination with exemestane to treat HR+ breast cancer underscores the potential HDACi and aromatase inhibitor combination therapies.

### 3.4. Combination with Immune Checkpoint Inhibitors

As described above, multiple preclinical studies provide a rationale for clinical trials based on the combination of HDAC inhibitors with immunotherapy. Yet, not many clinical trials have been conducted to date.

Forty-seven treatment-naïve patients with clear cell renal cell carcinoma were enrolled in a Phase I/II trial to receive entinostat in combination with high-dose IL2, the latter already being approved for treatment. The most common grade 3 or 4 adverse events were hypophosphatemia, lymphopenia and hypocalcemia. The efficacy analysis included 41 patients and identified 12 patients with PR and 3 with complete response (CR) equal to an ORR of 37%. Stable disease for over 6 months was achieved in 18 patients. Interestingly, one patient initially experienced progressive disease after receiving two cycles, but after discontinuation showed disease stabilization for over 3 years with ongoing slow reduction in lung nodules without any further therapy. Biopsies after the first cycles revealed increased CD8+ tumor infiltrating lymphocytes and decreased Tregs in patients with partial response or prolonged stable disease [237].

Another study enrolled 33 NSCLC patients, including 24 patients who had progressed on prior ICI therapy, for combined treatment with pembrolizumab and vorinostat. No dose limiting toxicities were observed. Out of six ICI-naïve patients, one showed PR and four stable disease (SD). Intriguingly, one patient with ICI refractory disease showed PR lasting for 12 months. In total, PR was observed in three patients and SD in 11 patients. No significant correlation was found between peripheral blood MDSCs and response rates upon treatment, probably because of the small sample number. Patients with NSCLC showed higher levels of MDSCs than heathy donors [238].

Pembrolizumab and vorinostat were also combined in a Phase II trial for HNSCC and salivary gland cancer, with 25 patients of each entity. In HNSCC, eight patients showed partial responses and five presented with stable disease. In salivary gland tumors, the response rates were lower with four partial responses and 14 patients with stable disease, among those four patients showed ongoing responses. By meeting the primary endpoints, the study showed encouraging results for further investigation of this drug combination [239]. The ENCORE 601 trial addressed the combination of entinostat and pembrolizumab in anti-PD-(L)1-resistant/refractory NSCLC patients. Out of 76 patients, 71 were evaluable for efficacy analysis. ORR was only 9.2% and therefore did not show a sufficient benefit to these patients. Out of the seven responders, five were negative for PD-L1 expression. In comparison with nonresponders, patients with longer PFSs presented with higher levels of classical monocytes [240].

More trials are underway for examining this treatment combination and identifying a suitable biomarker. For example, the INFORM2 NivEnt trial investigates the effects of nivolumab + entinostat in children and adolescents with high-risk malignancies. This Phase II basket trial includes four categories: high mutational load, high PD-L1 expression, high-level MYC amplified tumors and tumors with none of these characteristics [241]. Biomarker-driven studies may identify patients most likely to benefit from these drugs.

## 4. Bifunctional HDAC Inhibitors for Cancer Therapy

### 4.1. Multitarget Drugs: Advantages and Disadvantages

The use of drug combinations can provide increased efficacy by targeting additional disease-related pathways, can mitigate side effects and/or reverse drug resistance by blocking specific mechanisms of resistance [242]. Due to the multifactorial nature of tumorigenesis, the concept of polypharmacy is well-established in cancer therapy as it allows for the simultaneous interruption of different processes in order to arrest cell proliferation or induce apoptosis. To some extent, the effects of engaging multiple drug targets can be anticipated, but drug–drug interactions, whether beneficial or harmful, that depend on the respective pharmacokinetic behavior or yet undiscovered mechanisms, are rather unpredictable [243]. Desirable additive or synergistic effects of drug combinations may thus come at the risk of inducing adverse effects through unwanted drug–drug interactions and off-target activities that may ultimately impair patient compliance [242]. To tackle such complications, reducing the number of medications to a minimum has become a priority.

Polypharmacology is an emerging discipline in the field of drug discovery and aims at minimizing the downsides of polypharmacy by designing single drugs that are capable of interacting with multiple targets. It has long been observed that the efficacies of some established market drugs emanate from additional and often serendipitous modes of action that had not been considered during the initial development process [244,245]. In fact, Anighoro et al. point out that this is by no means a rare phenomenon but, on the contrary, a common characteristic among late clinical candidates which may even be considered as a contributing factor to the drugs’ success in the preclinical and clinical evaluation processes [242]. Elaborate efforts to detect unexpected or harmful off-target activities have indeed become a part of the drug development process but fail to provide a complete picture so that, for some drugs, the entirety of biological involvements remains enigmatic to date [242]. In some cases, however, careful profiling provided valuable information on synergistic or additional drug targets which could then be considered in the rational design process [245]. Thus, the interest in polypharmacology has grown over recent years. Ramsay et al. calculated that nearly every fifth drug approved by the FDA between 2015 and 2017 could be classified as a multitarget drug [246].

This clear trend toward multitarget ligands instead of combination drugs is further justified by the considerable reduction in cost and effort throughout the preclinical development process and the following clinical trials. Being subjected to only one pharmacokinetic process, multitarget ligands further guarantee the simultaneous presence in the designated tissues whereas the target-delivery of combination drugs might be deferred unless complicated dosing schedules adhere to [242]. One clear drawback of addressing multiple targets by administering a single drug might be the balancing of the doses required in each site; however, it has been suggested that untypically low doses of dual ligand drugs in synergistic targets suffice to elicit the desired efficacy [245,247]. Due to fewer interactions with healthy tissue, such low drug doses are moreover presumed to bear a reduced risk of side effects [245,247].

Considering the highly distinguished shapes of biological targets, it is clear that designing selective drugs for multiple targets is a challenge. In 2019, Merk and colleagues classified the different types of multitarget ligands as linked (Figure 2A), fused (Figure 2B) or merged (Figure 2C) pharmacophores [245].

This model is in accordance with an earlier overview provided by Morphy et al. who further divided the linked pharmacophores into cleavable and noncleavable conjugates [244]. In contrast, the simple method of functional group interchange suggested by de Lera and Ganesan rarely comes along with increased affinity to an additional target but may be successful if the respective groups are highly similar, such as carboxylic acids and hydroxamates [248].

Choosing between the different types of multitarget ligands to achieve specific target combinations generally requires excellent knowledge of structural characteristics of each binding site as well as careful design in order to meet the criteria for drug-likeness. Owing to their low molecular weight, however, merged dual ligands that match several binding sites with only one pharmacophore are considered to be particularly advantageous and several FDA-approved drugs and clinical candidates are already based on this concept [245,248].

### 4.2. Kinase Inhibiting HDACis

With regard to merged multitarget ligands bearing HDACi moieties, there is an overwhelming prevalence of dual HDAC/kinase inhibitors on both clinical and preclinical stages [211,249,250]. The structure of fimepinostat matches the HDAC pharmacophore model with a hydroxamate ZBG, a pyrimidine-linker, and a cap group inspired by the phosphoinositide 3-kinase (PI3K) inhibitor pictilisib [210,248]. Functioning as an inhibitor of PI3Kα and HDACs 1–3 and 10, the compound is currently undergoing Phase II trials against lymphomas and solid tumors. In another attempt from Curis, Inc. (Lexington, MA, USA), merging of the characteristic alkine moiety of the tyrosine kinase inhibitor erlotinib and an aliphatic HDAC linker bound to a hydroxamate ZBG provided the dual EGFR/HDAC inhibitor CUDC-101 (Figure 3) which has become a Phase I candidate for the treatment of solid cancers [204,211,248].

At the preclinical level, research on dual kinase/HDAC inhibitors is quickly evolving and compounds addressing receptor tyrosine kinases (RTKs), phosphoinositide 3-kinases (PI3Ks), the proto-oncogene tyrosine-protein kinase Src (c-Srcs), cyclin-dependent kinases (CDKs), and janus kinases (JAKs) have been presented. A detailed discussion of the progress in this field is beyond the scope of this article but has been extensively reviewed elsewhere [211,248,249,250,251,252,253].

### 4.3. DNA Damaging HDACis

Outside the field of kinase inhibition, one promising dual agent is tinostamustine (Figure 4). Designed as a hybrid of the pan-HDACi vorinostat and the alkylating agent bendamustine, tinostamustine acts as a potent HDACi with DNA-alkylating properties and has entered Phase I/II trials against lung cancer, brain tumors and hematological malignancies. Other analogues based on nitrogen mustard drugs were designed by Yuan and coworkers and have been investigated at the preclinical level [253,254,255]. The chlorambucil/vorinostat hybrid vorambucil outmatched its parent compounds in terms of both HDAC inhibition and antiproliferative potential in four cancer cell lines [254].

Chlordinaline, on the other hand, features the aminoanilide-based HDAC binding site of tacedinaline (CI-994) attached to the chlorambucil scaffold and displays moderate, but HDAC3-preferential inhibition and promising DNA damaging properties in vitro [255]. In another recent study, Sinatra et al. described a series of temozolomide/HDACi and chlorambucil/HDACi hybrids [256]. The most promising hybrid, compound 3n, a chimeric compound based on the pharmacophores of chlorambucil and panobinostat, displayed improved anticancer properties compared to the sum of the activities of the respective control compounds alone, indicating a superadditive effect [256].

### 4.4. HDAC-LSD1 Inhibitor

While all the aforementioned dual inhibitors were designed as such, there are also compounds whose actual dual activities emerged as a surprise. One example of such serendipity is the resminostat analogue domatinostat [248,257]. Originally believed to impair the function of the epigenetic eraser lysine-specific demethylase 1 (LSD1) which participates in the CoREST complex formation alongside HDACs 1 and 2, the class I HDACi unexpectedly turned out to inhibit tubulin polymerisation as well. Ongoing clinical trials in Phases I and II investigate its efficacy in hematological and gastrointestinal cancers [258,259].

Further research on dual LSD1/HDAC inhibitors yielded vorinostat derivative 7, featuring a tranylcypromine cap group that effectively inhibited both targets [260]—see Figure 5. By merging the original tranylcypromine group with entinostat, Kalin et al. developed the class I-selective HDAC/LSD1 inhibitor corin which inhibited the CoREST complex and reduced tumor growth in a melanoma mouse xenograft model [261].

### 4.5. Other Emerging Targets

In addition to the HDAC-involving synergisms explored in the clinic, dual HDAC inhibitors are thought to be useful for a multitude of additional targets and several prototypes have yet been reported in the literature (Figure 6). Through inhibiting HDACs, it is possible to maintain a relaxed chromatin structure which can be exploited to facilitate the access of DNA for topoisomerases [253]. Responsible for uncoiling the DNA superhelix by breaking and ligating single strands (topoisomerases I) or double strands (topoisomerases II), topoisomerases are crucial for replication and transcription so that inhibition results in arrested cell proliferation [253].

One class of compounds entertaining this mode of action are anthracyclines such as daunorubicin and its analogue doxorubicin. Intending to combine the activities of daunorubicin and the pan-HDACi vorinostat, Oyelere and coworkers introduced a small set of merged dual ligands of which compound 7 was singled out as a hit compound with promising cytotoxicity against different solid tumor cell lines [262]. In inhibition assays, compound 7 was observed to impair the activity of both HDACs and topoisomerase II at similar levels as vorinostat and daunorubicin, respectively [262]. Following this work, alternative structures based on the camptothecin, acridine and podophyllotoxin scaffolds have been reported [253,262,263,264,265].

While most attempts seek to benefit from the histone deacetylase activity of class I HDACs, it is noteworthy that recent in vitro combination studies also suggested promising synergistic effects of anthracyclines and selective HDAC6is [266,267].

Being part of the epigenetic network, HDACs function as erasers and happen to operate on molecular pathways and targets that are also served by the four bromo- and extra-terminal domain (BET) proteins (BRD2, BRD3, BRD4, BRDT) which recognize acetylated histones and are thus are classified as epigenetic readers [253]. Because of their presumed association with superenhancers that are suspected to boost cancer progression by assembling transcription factors near oncogenes, it was hypothesized that the most promising synergism concerns HDAC1, HDAC2 and particularly BRD4, which could be inhibited to disturb the transcriptional machinery of superenhancers [253,268,269].

Since the first report of DUAL946 by Atkinson et al. in 2014, several examples of dual BRD4/HDAC inhibitors have been disclosed [248,253,268,270,271]. Most recently, He et al. merged the first BET inhibitor (+)-JQ1 and the phenyl linker and hydroxamate ZBG of HDACi into their hit compound, 13a, which demonstrated superior antitumor activity than its parent compounds in a Capan-1 human pancreatic cancer xenograft model [272].

Through its function as a chaperone that assists in protein folding, Hsp90 is of particular importance for proliferating cells that rely on high protein expression. Interestingly, it has been observed that Hsp90 is activated by deacetylation and thus controlled by HDAC6 [30]. The concurrent application of HDACis is therefore presumed to increase the effect of Hsp90 inhibitors and can be further utilized to overcome acquired resistance [273].

Considering that the pharmacophores for HDAC6i and Hsp90 inhibitors show little similarity, the design of dual ligands appears to be particularly challenging, but nevertheless, some examples have been introduced over recent years [273]. Replacing the phenyl cap group of vorinostat by the resorcinol moiety present in the Phase II Hsp90 inhibitors luminespib and onalespib, recently yielded compound 12, which inhibited both targets and induced apoptosis in lung cancer cells [274]. Another group studying the resorcinol scaffold chose to discard the indoline motif of onalespib in favor of a phenyl group and achieved similar effects [275]. As expected, both compounds induced the upregulation of Hsp70 and the degradation of Hsp90 client proteins [274,275].

### 4.6. PROTACs

The so-called PROteolysis-TArgeting Chimeras (PROTACs) are emerging therapeutic modalities in modern drug discovery. PROTACs are bifunctional small molecules consisting of an E3 ubiquitin ligase recognition motif and a ligand for the protein of interest (POI) connected by a suitable linker. Due to their bifunctional nature, they can act as proximity inducers and catalyze the formation of an E3 ligase:PROTAC:POI ternary complex. As a consequence, they are capable of hijacking the cellular protein degradation system by inducing polyubiquitinylation and subsequent proteasomal degradation of the POI. This approach might offer significant advantages over classical inhibition strategies using small molecules including (1) a catalytic mode of action, (2) the avoidance of resistance due to upregulation of the POI, (3) the possibility of drugging currently undruggable targets by targeted protein degradation and (4) the removal of all possible functions (i.e., enzymatic, scaffolding, regulatory, etc.) of the POI [276]. Very recently, ARV-110 (an androgen receptor degrader) and ARV-471 (an estrogen receptor degrader) have entered clinical Phase I trials as the first PROTACs and have shown promising early data in terms of tolerability, safety and efficacy, thus highlighting the potential of targeted protein degradation to combat cancer [276].

The field of histone deacetylase degraders (HDAC PROTACs) is a very young research area. In 2018, Schiedel et al. disclosed the first degrader of the NAD^+^-dependent histone deacetylase sirtuin 2 [277]. The first degrader of classical Zn^2+^-dependent HDACs was reported by Yang et al. in 2018 [278]. The most promising PROTAC from this series (dHDAC6, Figure 7) turned out to be an efficient and selective degrader of HDAC6. The selective degradation of HDAC6 over HDACs 1, 2 and 4 is somewhat surprising, since the degrader design was based on the unselective HDACi crebinostat (Figure 7) and pomalidomide as a ligand for the E3 ubiquitin ligase cereblon (CRBN). Whether the cellular localization of HDAC6 or a more efficient ternary complex formation is responsible for the selective degradation of HDAC6 is unknown. While dHDAC6 demonstrated efficient degradation of HDAC6 in MCF-7 breast cancer cells, the multiple myeloma cell line MM.1S was more sensitive to dHDAC6 in regard to degradation of HDAC6 [278,279].

Since the initial report by Yang et al., several other HDAC degraders have been disclosed, including selective HDAC6 PROTACs [256,279,280,281,282,283], class I-selective PROTACs [284,285,286] and HDAC3-selective degraders [287]. Although most work has focused on hematological cancers so far, there are some encouraging results indicating that HDAC PROTACs could be an innovative new option for the treatment of solid cancers. For instance, the Rao lab recently published a series of HDAC6-selective PROTACs based on the selective HDAC6 inhibitor nexturastat. The CRBN-recruiting PROTAC NH2 (Figure 7) emerged as the best degrader from this study, demonstrating potent and selective degradation of HDAC6 in multiple cancer cell lines including HeLa and MDA-MB-231 cells [281].

In another recent study, Smalley et al. designed and synthesized a series of class I-selective HDAC degraders [284]. To this end, the class I-selective 2-aminoanilide CI-994 was linked via alkyl linkers to either pomalidomide as a CRBN ligand or VH032 as a Von Hippel-Lindau (VHL) E3 ligase ligand. The VH032-based PROTAC 4 (Figure 7), the most efficient degrader from this series, demonstrated at least 50% degradation of HDACs 1, 2 and 3 in HCT116 colon cancer cells at a concentration of 1 µM. After a 48 h treatment, PROTAC 4 exhibited comparable effects on HCT116 cell viability as its parent compound CI-994 [284].

By utilizing a benzoylhydrazide-based ZBG, Xiao et al. reported a new type of HDAC PROTAC capable of degrading HDAC3 in a potent and selective manner [287]. The most promising PROTAC XZ9002 (Figure 7) dose-dependently induced HDAC3 degradation in the triple negative breast cancer cell line MDA-MB-468, with a DC_50_ value (half-degrading concentration) of 42 nM. Furthermore, XZ9002 (Figure 7) efficiently suppressed clonogenic growth of T47D, HCC1143, MDA-MB-468 and BT549 breast cancer cells [287].

## 5. Outlook

With the first HDAC degrader being reported in 2018, this research area is too young to judge whether HDAC degradation by HDAC PROTACs will be able to provide improved anticancer effects compared to classical HDAC inhibition by HDACis. However, it is reasonable to assume that the cellular effects of degraders and inhibitors will be different. PROTACs are expected to achieve high cellular potency due to their catalytic modes of action and might have a longer duration of action, since the latter depends on the turnover rate of the protein target rather than on residence time of the inhibitor [288]. Furthermore, PROTACs will have different effects on their targets than inhibitors. For instance, the targeted degradation of HDAC6 eliminates both catalytic domains as well as the ubiquitin-binding domain, whereas the inhibition of HDAC6 by a selective HDAC6i only blocks the second catalytic domain of HDAC6. HDACs are often localized in multienzyme complexes. Thus, the formation of the E3 ligase:PROTAC:HDAC ternary complex might also lead to polyubiquitinylation of other proteins involved in the multienzyme complex by a so-called “bystander ubiquitination” [289]. This could result in improved anticancer effects, but also in additional side effects. Targeted degradation of HDACs is still in its infancy and more research on the preclinical and clinical levels is needed to clarify whether HDAC PROTACs can demonstrate efficacy and safety. Furthermore, gene-editing approaches could offer new avenues to control the acetylation statuses of proteins. For instance, Kwon et al. described the dCas9-HDAC3 system as a unique addition to the CRISPR-dCas9 epigenome-editing toolbox [290]. The method could provide a new way to modulate the histone deacetylation of genomic loci associated with various developmental and disease states. However, further work is needed to fully explore the potential of this approach.

From the present synopsis on the status of HDACis in cancer therapy for solid tumors, one can draw the conclusion that these compounds will most likely fulfill their full potential only in combination therapies, which will help to overcome some shortcomings of HDACis. These combinations should be based on the still better understanding of the mechanisms of HDACi-promoted anticancer effects. Of particular relevance will be the definition of therapeutic biomarkers, for predicting the susceptibility of cancer cells to distinct, HDAC subtype-specific HDACis or dual inhibitors, targeting HDAC function plus additional oncogenic signaling pathways. Thus, it remains a critical endeavor to unravel the diverse mechanisms induced by HDAC inhibition.

Owing to the heterogeneity of cancer, there will likely be no ideal one-fits-all combination of an HDACi plus another drug for a given tumor entity or even for all cancer types. Rather, individualized therapeutic combinations will have to be delineated, incorporating the molecular signatures of a patient’s tumor and thus leading to personalized medicine.

## 6. Conclusions

HDACis show enormous potential for attenuating tumor growth and provoking cell death in a multitude of solid cancer entities. Our review points out the influence of epigenetic changes upon HDACi treatment on various essential signaling pathways. The overall importance of HDACis is also highlighted by the large number of current clinical studies on combination therapies. In this regard, the recent development of bifunctional inhibitors or new molecules such as PROTACs may be particularly promising. In summary, HDAC inhibition may have a substantial role in future targeted therapy of solid tumors.

## Figures and Tables

**Figure 1 cancers-13-00634-f001:**
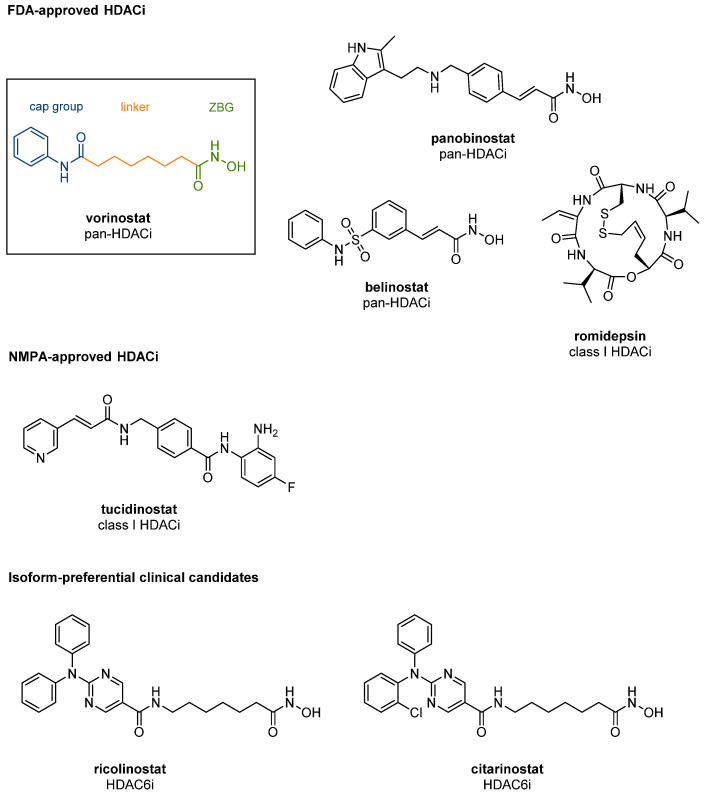
FDA/NMPA-approved histone deacetylase (HDAC) inhibitor (HDACi) and isoform-preferential clinical candidates.

**Figure 2 cancers-13-00634-f002:**
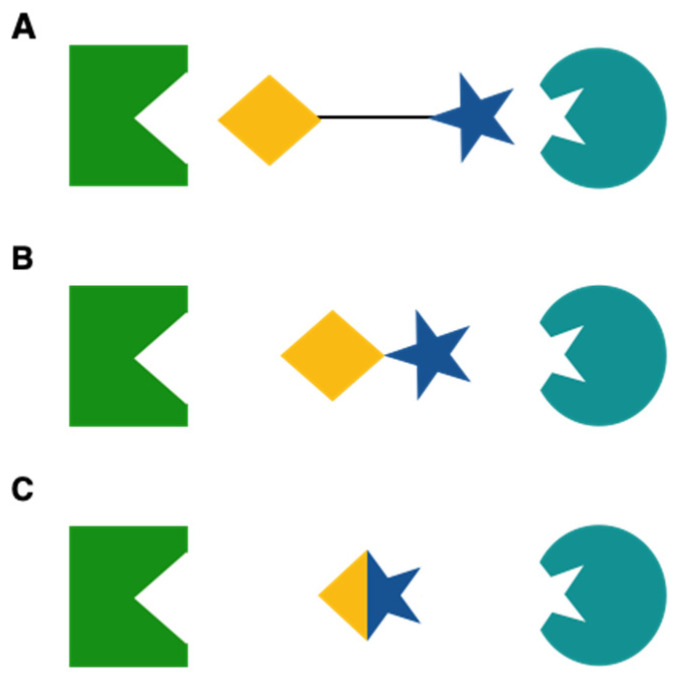
Concept of linked (**A**), fused (**B**) or merged (**C**) pharmacophores.

**Figure 3 cancers-13-00634-f003:**
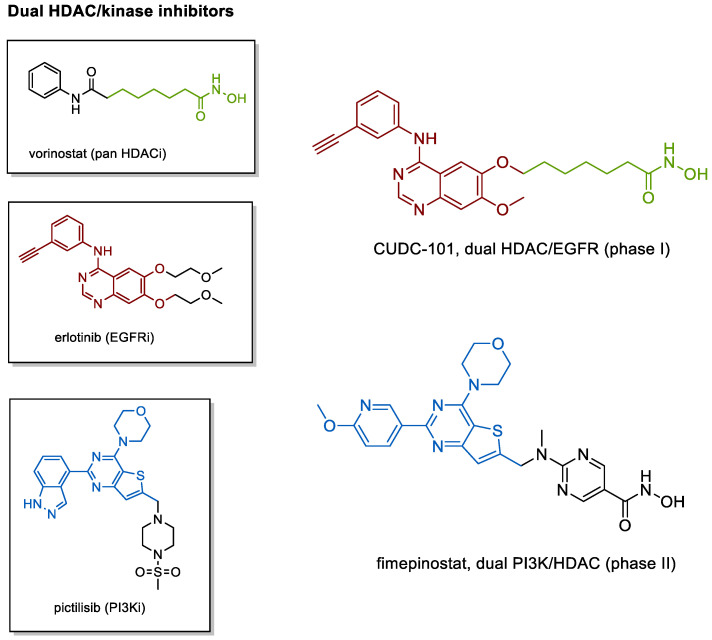
Chemical structures of the dual HDAC/kinase inhibitors CUDC-101 and fimepinostat.

**Figure 4 cancers-13-00634-f004:**
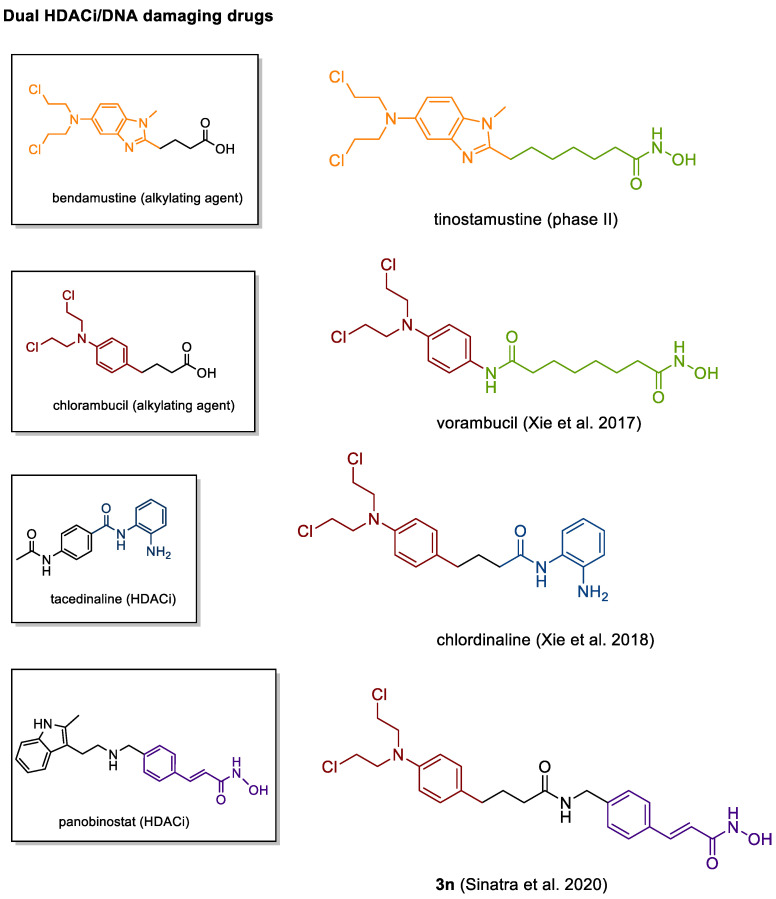
HDACi with DNA-alkylating properties.

**Figure 5 cancers-13-00634-f005:**
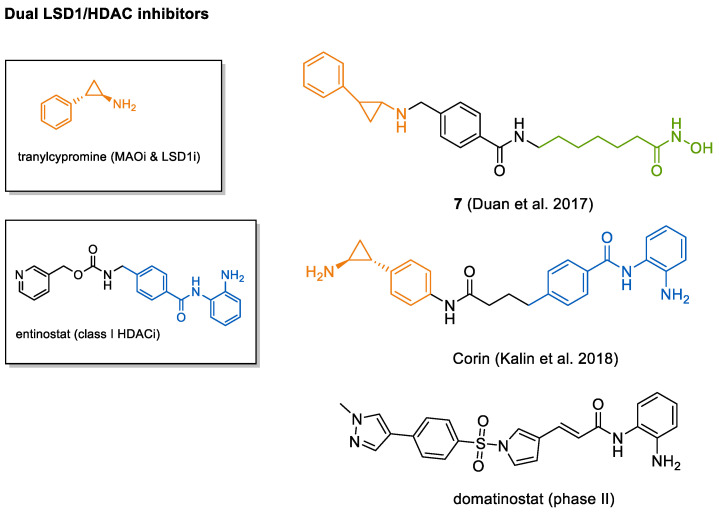
Selected dual lysine-specific demethylase 1 (LSD1)/HDAC inhibitors.

**Figure 6 cancers-13-00634-f006:**
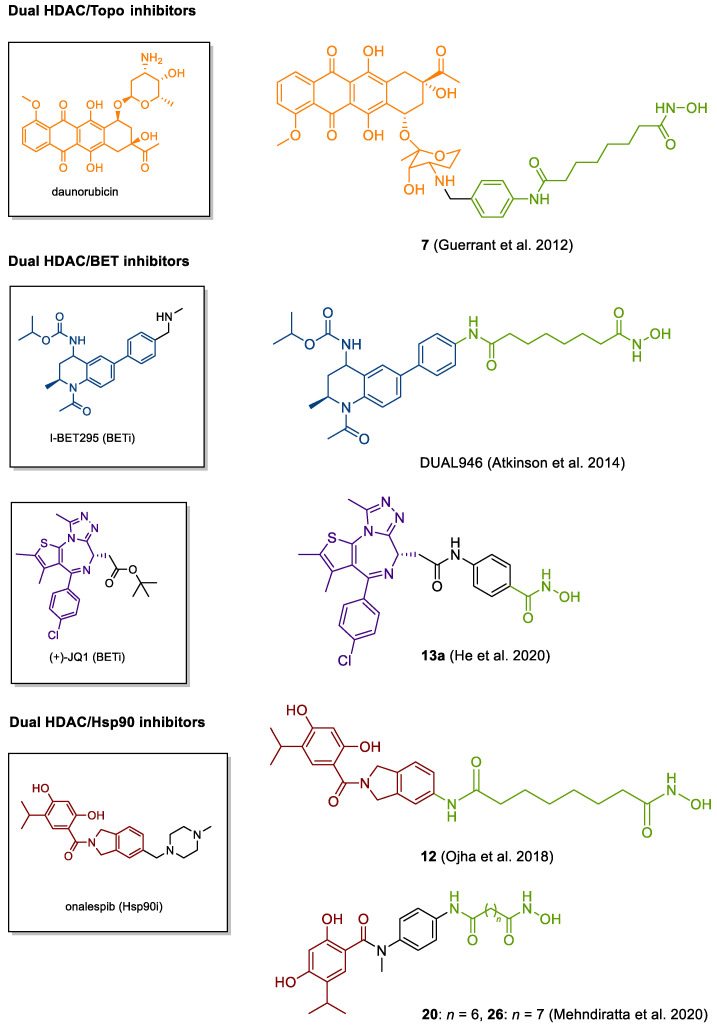
Selected examples of bifunctional HDAC inhibitors.

**Figure 7 cancers-13-00634-f007:**
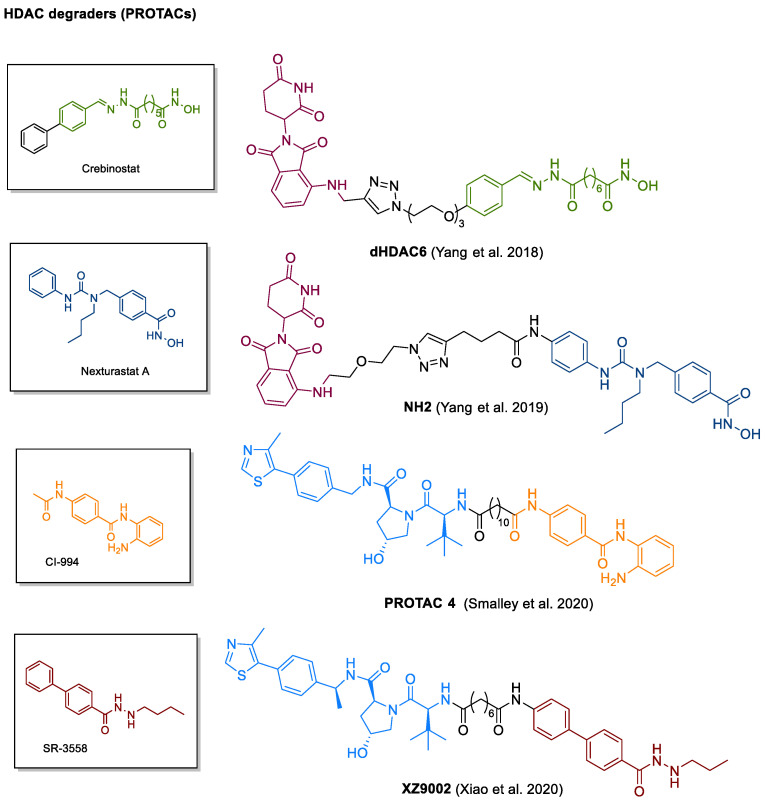
Selected histone deacetylase degraders (PROteolysis-TArgeting Chimeras; PROTACs).

## Data Availability

The data discussed in this study are publicly available and can be found under the references cited in the text.

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
