# Peer review of "Anticancer Therapy with HDAC Inhibitors: Mechanism-Based Combination Strategies and Future Perspectives"

_cancers, 2021, doi:10.3390/cancers13040634_

Round 1

Reviewer 1 Report

This is a very thorough review from Jenke and colleagues summarising the role of HDACs and HDAC inhibitors (HDACi) in the treatment of various forms of cancer. The manuscript falls into roughly 3 areas: i) a review of the enzymes and inhibitors and potential mechanisms for their anti-proliferative and pro-apoptotic behaviour; ii) the use of HDACi in combinations with other therapeutics and finally, iii) Bi-functional HDACi. The article is beautifully written and well-referenced. It will make a strong contribution to the field. The summary of clinical trials with HDACi in combination with other therapeutics is impressive and will function as a useful short-cut for many researcher working within the HDAC community. The review is at it's best in section 3, I thought the update and bifunctional HDACi was excellent.

The manuscript is accurate and well referenced, I have very few comments in this regard (see below). My main comments are stylistic and structural. I was surprised that the authors chose to leave out any discussion of HDACi in the treatment of CTCL, when this is the most obvious clinical success story for HDACi. This seems like an odd omission when many of the examples given are either pre-clinical or clinical trials. I would be curious to hear the authors opinion as to why CTCL stands-out as a cell type / malignancy that responds better HDACi than other cancer types. My second structural comment, is in relation to the mechanistic section. These are well written, but I'm surprised that there is no section on DNA damage and/or genome stability given the major roles that HDACs play, particularly class-I HDACs. HDAC1/2 has reasonably well characterised roles in double-strand break repair e.g. Miller et al., https://www.nature.com/articles/nsmb.1899.pdf , reviewed in https://www.tandfonline.com/doi/full/10.1080/15384101.2015.1042634

In my opinion, the likeliest explanation for the cytoxicity of most HDACi is due to direct effects on DNA synthesis, damage repair and consequently genome stability via their role in modulating chromatin structure and histone deposition during cell cycle. Instead, the authors go to some lengths to discuss autophagy, senescence, hormone signalling, etc., which to my mind are all secondary effects. Anyway, that's my opinion, and the authors are welcome to theirs, but I would appreciate a new section covering some of the above. My final comment is more of a generalised question, to the authors and the field at large. How do much of the potential mechanisms described herein fit with recent data from the Chaudhary lab showing that the vast majority of Lysine-acetylation occurs in histones..? https://www.nature.com/articles/s41467-019-09024-0

Given this data, are we still to believe that p53 is still a physiologically relevant for HDACs..? Food for thought.

Minor comments:

A characteristic mutation present in the active sites of all class IIa isoforms is the replacement of a catalytically crucial tyrosine by a histidine residue which can rotate to open the lower pocket [48] – This is not the primary reference for the amino acid change within HDACIIa enzymes. Lahm et al 2007, would be better

https://doi.org/10.1073/pnas.0706487104

...serving as an ubiquitin-binding domain (HDAC6 UBD) [42]. This is not the primary reference for the HDAC6 UBD. Either of...

Seigneurin-Berny wt al., 2001 - DOI:10.1128/MCB.21.23.8035-8044.2001

or

Hook et al., 2002 - https://doi.org/10.1073/pnas.172511699

Would be better.

Author Response

Response to Reviewer 1 Comments

Comment: This is a very thorough review … The article is beautifully written and well-referenced. It will make a strong contribution to the field. The summary of clinical trials with HDACi in combination with other therapeutics is impressive and will function as a useful short-cut for many researcher working within the HDAC community. The review is at it's best in section 3, I thought the update and bifunctional HDACi was excellent.

The manuscript is accurate and well referenced, I have very few comments in this regard (see below). My main comments are stylistic and structural.

Reply: We are happy about the Reviewer’s very positive evaluation of our article. In the revised version, his / her specific comments have been addressed as follows:

Comment: I was surprised that the authors chose to leave out any discussion of HDACi in the treatment of CTCL, when this is the most obvious clinical success story for HDACi. This seems like an odd omission when many of the examples given are either pre-clinical or clinical trials. I would be curious to hear the authors opinion as to why CTCL stands-out as a cell type / malignancy that responds better HDACi than other cancer types.

Reply: Indeed, CTLC stands out as an entity, responding extremely well to HDAC inhibition. One possible explanation points towards the unique DNA architecture causing distinct DNA accessibility features and subsequent transcription factor activation rather than relying on driver mutations like in solid tumors.

Following the Reviewer’s suggestion, we have made additions to the manuscript as follows:

“Of note, the application of HDAC inhibitors proved to be mainly useful in hematological disease [18]. For example, HDAC inhibitor therapy has been shown as an efficacious option in cutaneous T cell lymphoma (CTCL) [19]. Approximately 30% of CTCL pa-tients respond to HDACi treatment, which might be attributable to genetic changes for example in adhesion pathways that can also be found in solid tumors [20]. However, out-standing responses in CTCL are more likely due to distinct DNA accessibility features. For example, one observed cluster showed increased access to the HDAC9 locus, responsible for Foxp3-dependent suppression in T regulatory cells. This response to HDACi was highly dependent on transcription factor enrichment of CTCF, affecting overall chromatin structure. In contrast to solid tumors, the unique 3D DNA structure of CTCL rather than oncogenic mutations might be accountable for effective monotherapy with HDACi in this entity [21].

We appreciate the Reviewer’s comment, prompting us to make this important addition.

Comment: My second structural comment is in relation to the mechanistic section. These are well written, but I'm surprised that there is no section on DNA damage and/or genome stability given the major roles that HDACs play, particularly class-I HDACs. HDAC1/2 has reasonably well characterised roles in double-strand break repair e.g. Miller et al., https://www.nature.com/articles/nsmb.1899.pdf , reviewed in https://www.tandfonline.com/doi/full/10.1080/15384101.2015.1042634

Reply: The Reviewer is correct in pointing out that genome stability / DNA damage are important aspects in this context. In the revised manuscript, we have added a new sub-chapter on this topic, chapter 2.4 on DNA damage, which also covers the important findings of Miller et al. In fact, we feel that this new section is a very useful complement of the review and are very grateful for suggesting this addition.

The impact of HDAC inhibitors on DNA integrity is another pivotal aspect with regard to cytotoxic responses upon treatment with these agents [37,121]. In some cases, the appearance of DNA lesions after exposure to HDAC inhibitors may be merely a consequence of their pro-apoptotic or cytotoxic effects. However, DNA damage may often represent an initial and critical molecular event that is directly responsible for the anticancer effects of HDAC inhibitors.

A better understanding of the still not fully resolved mechanisms behind the occurrence of DNA damage after HDAC inhibitor treatment is a central issue with regard to the therapeutic use of these substances and the development of novel inhibitors. In fact, HDAC inhibitors have been described as genotoxic or mutagenic agents in a number of reports in malignant [122–124] as well as in non-malignant cells [125–128]. From these findings, the question arises whether HDAC inhibitors have a carcinogenic potential, which would be especially relevant when considering their therapeutic use in younger patients and/or in non-oncological diseases. Although the DNA damaging effects of HDAC inhibitors have been found to be more pronounced in malignant than in non-malignant cells according to some reports [129,130], this issue clearly demands further clarification.

Regarding the mechanisms of DNA damage induced by HDAC inhibitors, two plausible explanations have been proposed: (1) the induction of oxidative stress by HDAC inhibitors and (2) the inhibition of the DNA repair machinery, with the subsequent accumulation of DNA lesions evoked by endogenous or exogenous mutagens.

The issue of oxidative stress induction by HDAC inhibitors has already been discussed above in the context of apoptosis induction (see 2.1). An involvement of reactive oxygen or nitrogen species in DNA damage related to HDAC inhibitors is suggested by the fact that an increase in markers of oxidative DNA stress has been associated with HDAC inhibitor treatment [131,132]. One important question to be addressed in this context is, whether oxidative stress elicited by HDAC inhibitors is a consequence of the enzyme inhibition by these agents or a result of reactive decomposition products of the inhibitor molecule itself. In the latter case, at least some genotoxic effects of HDAC inhibitors would be independent of HDACs and show significant differences, depending on the chemical structure of the inhibitor molecule. Of note, hydroxamic acid derivatives (also representing one important class of HDAC inhibitors) may under certain conditions give rise to isocyanates [133–135], which can directly or indirectly lead to DNA modifications. Moreover, hydroxamic acids may release nitrogen monoxide [136], which could be also responsible for an increase in oxidative stress. However, it is still unsettled whether these reactions also occur under physiological conditions in a living cell.

The second pathway, which has been proposed to result in an accumulation of DNA damage, is the interference with DNA repair mechanism by HDAC inhibitors [137–140]. This effect of HDAC inhibitors suggests a synergism upon combination with DNA damaging chemotherapeutics, providing a rationale for the generation of hybrid molecules with combined HDAC inhibiting and DNA alkylating properties as discussed below (see: 4.3). From the mechanistic point of view, impairment of DNA repair by HDAC inhibitors may be a consequence of (1) altered DNA architecture or (2) dysregulated expression or activity of DNA repair enzymes and/or DNA damage signaling.

With respect to the structural organization of DNA, histone modifications are one key element affecting the DNA access of mutagenic substances, but also of repair proteins. Thus, chromatin organization is of paramount importance for the integrity of the genome [141–143]. The next level, i.e., the HDAC-dependent regulation of the expression and function of the DNA repair machinery, is a highly complex issue, since virtually all types of DNA repair mechanisms are impacted by HDACs [140,144]. Thus, the in-depth discussion of DNA repair proteins that have been shown to be affected by HDAC inhibitors would go far beyond the scope of this review. However, two important aspects should be mentioned in brief.

Firstly, HDAC1 has been shown to directly stimulate oxoguanine glycosylase 1 (OGG1), a repair protein critically involved in base excision of oxidized guanine residues, whereas HDAC1 deficiency causes impairment of OGG1 activity [145]. Thus, an increase of 8-oxoguanine in DNA after HDAC inhibition could be the consequence of increased oxidative stress (see above) or impaired repair of this lesion. Therefore, the detection of 8-oxoguanine lesions after HDAC inhibitor treatment may be insufficient for proving that this agent per se augments oxidative stress.

Secondly, the repair of one of the most lethal DNA damages, the induction of DNA double strand breaks, is critically regulated by HDAC subtypes 1 and 2, which are directly recruited to DNA damage sites, whereas other HDAC isotypes such as HDAC3 are not involved in this process [146]. From this finding, HDAC1/2 subtype-specific inhibitors should be exceptionally well-suited for combination therapies with DNA damaging agents inducing double strand breaks.”

Comment: How do much of the potential mechanisms described herein fit with recent data from the Chaudhary lab showing that the vast majority of Lysine-acetylation occurs in histones..? https://www.nature.com/articles/s41467-019-09024-0

Reply: We are grateful to Reviewer #1 for mentioning this interesting work. Indeed, it fits very well into the observation that class I HDAC inhibitors, which primarily act on histones, are particularly promising drug candidates. We therefore included the information that the vast majority of lysine acetylation occurs at histones in our revised manuscript, and also cite this paper. The new and modified text reads as follows:

“The four class I HDACs were reported to act on histones where the vast majority of cellular lysine acetylation takes place. Moreover, even though class IIa HDACs might still play a part in the histone deacetylation process through complex formation with HDAC3, it is now questionable whether they exert any independent deacetylase activity [28,29].”

Comment: A characteristic mutation present in the active sites of all class IIa isoforms is the replacement of a catalytically crucial tyrosine by a histidine residue which can rotate to open the lower pocket [48] 

 => This is not the primary reference for the amino acid change within HDACIIa enzymes. Lahm et al 2007, would be better

Reply:  We agree with the Reviewer and replaced the reference.

Comment:   ...serving as an ubiquitin-binding domain (HDAC6 UBD) [42]. => This is not the primary reference for the HDAC6 UBD. Either of….- Seigneurin-Berny wt al., 2001 - DOI:10.1128/MCB.21.23.8035-8044.2001 or- Hook et al., 2002 - https://doi.org/10.1073/pnas.172511699 would be better.

Reply: Again, we thank the Reviewer 1 for the suggestions and included the reference Hook et al. 2002.

Reviewer 2 Report

It is a well-written comprehensive overview of anticancer activity of HDAC inhibitors (HDACi) describing their mechanisms of action and presenting their mechanism-based preclinical and clinical studies. The authors also present different therapeutic approaches including combinations of HDACi with other anticancer agents and as polypharmacology-based multitarget inhibitors, and new proteolysis-targeting chimeras (PROTACs). In general, it is a well-assembled review focusing on the use of HDACi in different ways to enhance their anticancer efficacies.

A few suggestion and comments for improvements are as follows:

  1. Several abbreviations /acronyms need to be defined at first usage. Some examples include,

    3-MA on line 235

    TNBC on line 237

    HCC on line 249

    VPA on line 282

    HNSCC and NSCLC on line 289

    SERMs on line 544

    PFS on line  573

    ORR on 578

  1. Inclusion of some relevant literature related to CUDC-907 such as PMID: 32459381; PMID: 33147762; PMID: 33049098; and PMID: 31218726.
  2. Targeted degradation of HDACs via the PROTACs approach is interesting. However, as the authors indicate, this approach needs more research to confirm its efficacy.                                                                  Interesting molecular studies in cancer research are going on using the gene-editing approach. This approach may have potential use in correcting the abnormal expression of HDACs. Again, this approach also needs more research to confirm its efficacy.

Author Response

Response to Reviewer 2 Comments

Reviewer 2

Comment: It is a well-written comprehensive overview of anticancer activity of HDAC inhibitors (HDACi) describing their mechanisms of action and presenting their mechanism-based preclinical and clinical studies. The authors also present different therapeutic approaches including combinations of HDACi with other anticancer agents and as polypharmacology-based multitarget inhibitors, and new proteolysis-targeting chimeras (PROTACs). In general, it is a well-assembled review focusing on the use of HDACi in different ways to enhance their anticancer efficacies. A few suggestion and comments for improvements are as follows …

Reply: We are happy about the Reviewer’s very positive evaluation of our article. In the revised version, his / her specific comments have been addressed as detailed below:

Comment: Several abbreviations /acronyms need to be defined at first usage. Some examples include, 3-MA on line 235

    TNBC on line 237

    HCC on line 249

    VPA on line 282

    HNSCC and NSCLC on line 289

    SERMs on line 544

    PFS on line  573

    ORR on 578

Reply: We agree with the Reviewer that this will help the reader who may not necessarily know all the above abbreviations / acronyms. They are now all defined in the text.

Comment: Inclusion of some relevant literature related to CUDC-907 such as PMID: 32459381; PMID: 33147762; PMID: 33049098; and PMID: 31218726.

Reply: We are grateful to the Reviewer for providing additional input and information. The above-mentioned literature has been included in our revision. The new text reads as follows:

“For example, proliferation of chronic lymphatic leukemia cells was inhibited via downregulation of anti-apoptotic Bcl-2 family members [185]. CUDC-907 evades drug resistance and also showed effects in myc-driven lymphoma cells [186]. Next to the increase in apoptosis in prostate cancer cells, CUDC-907 also mediated anti-proliferative effects through inducing DNA-damage [187]. Multiple entities have shown sensitivity towards fimepinostat, thus it is further evaluated in Phase I/II clinical trials [188].”

Comment: Targeted degradation of HDACs via the PROTACs approach is interesting. However, as the authors indicate, this approach needs more research to confirm its efficacy. Interesting molecular studies in cancer research are going on using the gene-editing approach. This approach may have potential use in correcting the abnormal expression of HDACs. Again, this approach also needs more research to confirm its efficacy.

Reply: We agree with Reviewer #2 that gene-editing approaches offer interesting new options to perform epigenome-editing. Following his/her comment, we updated the discussion and added the following new text.

“Furthermore, gene-editing approaches could offer new avenues to control the acetylation status of proteins. For instance, Kwon et al. described the dCas9-HDAC3 system as a unique addition to the CRISPR-dCas9 epigenome-editing toolbox. The method could provide a new way to modulate the histone deacetylation of genomic loci associated with various developmental and disease states. However, further work is needed to fully explore the potential of this approach.”
